# Learning Intuitive Policies Using Action Features

## Abstract

An unaddressed challenge in multi-agent coordination is to enable AI agents to exploit the semantic relationships between the features of actions and the features of observations. Humans take advantage of these relationships in highly intuitive ways. For instance, in the absence of a shared language, we might point to the object we desire or hold up our fingers to indicate how many objects we want. To address this challenge, we investigate the effect of network architecture on the propensity of learning algorithms to exploit these semantic relationships. Across a procedurally generated coordination task, we find that attention-based architectures that jointly process a featurized representation of observations and actions have a better inductive bias for learning intuitive policies. Through fine-grained evaluation and scenario analysis, we show that the resulting policies are human-interpretable. Moreover, such agents coordinate with people without training on any human data.

## 1 Introduction

Successful collaboration between agents requires coordination (Tomasello et al., 2005; Misyak et al., 2014; Kleiman-Weiner et al., 2016), which is challenging because coordinated strategies can be arbitrary (Lewis, 1969; Young, 1993; Lerer & Peysakhovich, 2018). A priori, one can neither deduce which side of the road to drive, nor what utterance to use to refer to ♡ (Pal et al., 2020). In these cases coordination can arise from actors best responding to what others are already doing—i.e., following a convention. For example, Americans drive on the right side of the road and say "heart" to refer to ♡ while Japanese drive on the left and say "shinzo". Yet in many situations prior conventions may not be available and agents may be faced with entirely novel situations or partners. In this work, we study ways that agents may learn to leverage semantic relations between observations and actions to coordinate with agents they have had no experience interacting with before.

Consider the shapes in Fig. 1. When asked to assign the names "Bouba" and "Kiki" to the two shapes, people name the jagged object "Kiki" and the curvy object "Bouba" (Köhler, 1929). This finding is robust across different linguistic communities and cultures and is even found in young children (Maurer et al., 2006). The causal explanation is that people match a "jaggedness"-feature and "curvey"-feature in both the visual and auditory data. Across the above cases, there seems to be a generalized mechanism for mapping the features of the person's action with the features of the action that the person

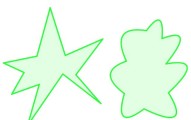

Figure 1: The "Bouba" (right) and "Kiki" (left) effect.

desires the other agent to take. In the absence of norms or conventions, people may minimize the distance between these features when making a choice. This basic form of feature utilization in humans predates verbal behavior (Tomasello et al., 2007), and this capability has been hypothesized as a key predecessor to more sophisticated language development and acquisition (Tomasello et al., 2005). Modeling these capacities is key for building machines that can robustly coordinate with other agents and with people (Kleiman-Weiner et al., 2016; Dafoe et al., 2020).

Might this general mechanism emerge through multi-agent reinforcement learning across a range of tasks? As we will show, reinforcement learning agents naively trained with self-play fail to learn to coordinate even in these obvious ways. Instead, they develop arbitrary private languages that are

uninterpretable to both the *same* models trained with a different random seed and to human partners (Hu et al., 2020). For instance, in the examples above, they would be equally likely to wave a red-hat to hint they want strawberries as they would to indicate that they want blueberries.

Unfortunately, developing an inductive bias that might take into account these correspondences is not straightforward because describing the kind of abstract knowledge that these agents lack in closed form is challenging. Rather than attempting to do so, we take a *learning-based* approach. Our aim is to build an agent with the capacity to develop these kinds of abstract correspondences during self-play, such that it can robustly succeed during *cross play*, a setting in which a model is paired with a partner (human or AI) that it did not train with.

Toward this end, we extend the Dec-POMDP formalism to allow actions and observations to be represented using shared features and design a human-interpretable environment for studying coordination with these enrichments. Using this formalism, we examine the inductive bias of a collection of five network architectures—two feedforward based and three attention based—in procedurally generated coordination tasks. Our main contribution is finding that a self-attention architecture that takes *both* the action and observations as input has a strong inductive bias toward using the relationship between actions and observations in intuitive ways. This inductive bias manifests in:1) High intra-algorithm cross-play scores compared to both other architecures and to algorithms specifically designed to maximize intra-algorithm cross play; 2) Sophisticated human-like coordination patterns that exploit mutual exclusivity and implicature—two well-known phenomena studied in cognitive science (Markman & Wachtel, 1988; Grice, 1975); 3) Human-level performance at ad-hoc coordinating with humans. We hypothesize that the success of this architecture can be attributed to the fact that it processes observation features and action features using the same weights. Our finding suggests that this kind of attention architecture is the most sensible starting point for learning intuitive policies in settings in which action features play an important role.

## 2 BACKGROUND AND RELATED WORK

**Dec-POMDPs.** We start with decentralized partially observable Markov decision processes (Dec-POMDPs) to formalize our setting (Nair et al., 2003). In a Dec-POMDP, each player $i$ receives an observation $\Omega^i(s) \in \mathcal{O}^i$ generated by the underlying state $s$, and takes action $a^i \in \mathcal{A}^i$. Players receive a common reward $R(s, a)$ and the state transitions according to the function $\mathcal{T}(s, a)$. The historical trajectory is $\tau = (s_1, a_1, \ldots, a_{t-1}, s_t)$. Player $i$'s action-observation history (AOH) is denoted as $\tau_t^i = (\Omega^i(s_1), a_1^i, \ldots, a_{t-1}^i, \Omega^i(s_t))$. The policy for player $i$ takes as input an AOH and outputs a distribution over actions, denoted by $\pi^i(a^i \mid \tau_t^i)$. The joint policy is denoted by $\pi$.

**MARL and Coordination.** The standard paradigm for training multi-agent reinforcement learning (MARL) agents in Dec-POMDPs is self play (SP). However, the failure of such policies to achieve high reward when evaluated in cross play (XP) is well documented. Carroll et al. (2019) used grid-world MDPs to show that both SP and population-based training fail when paired with human collaborators. Bard et al. (2019); Hu et al. (2020) showed that agents perform significantly worse when paired with independently trained agents than they do at training time in Hanabi, even though the agents are trained under identical circumstances. This drop in XP performance directly results in poor human-AI coordination, as shown in (Hu et al., 2020). Lanctot et al. (2017) found similar qualitative XP results in a partially-cooperative laser tag game.

To address this issue, Hu et al. (2020) introduced a setting in which the goal is to maximize the XP returns of independently trained agents using the same algorithm. We call this setting *intra-algorithm cross play* (intra-AXP). Hu et al. (2020) argue that high intra-AXP is necessary for successful coordination with humans: If agents trained from independent runs or random seeds using the same algorithm cannot coordinate well with each other, it is unlikely they will be able to coordinate with agents with different model architectures, not to mention humans. However, while there has been recent progress in developing algorithms that achieve high intra-AXP scores in some settings Hu et al. (2020; 2021), this success does not carry over to settings in which the correspondence between actions and observations is important for coordination, as we will show.

Beyond performing well in intra-AXP, a more ambitious goal is to perform well with agents that are externally determined, such as humans, and not observed during training time. This setting has been referred to both as ad-hoc coordination (Stone et al., 2010; Barrett et al., 2011) and zero-shot

coordination (Hu et al., 2020; Strouse et al., 2021). However, both terms carry some ambiguity, as ad-hoc coordination is sometimes evaluated in a setting in which the externally determined partners are known at training time (Stone et al., 2010) and zero-shot coordination is sometimes used to refer to the setting in which the goal is to maximize intra-AXP (Hu et al., 2020). We use ad-hoc coordination to refer to this setting.

**Dot-Product Attention.** As we will see in our experiments, one way to leverage the correspondences between action features and observation features is by using attention mechanisms (Vaswani et al., 2017; Bahdanau et al., 2015; Xu et al., 2016). Given a set of input vectors $(x_1, ..., x_m)$, self-attention uses three weight matrices $(Q, K, V)$ to obtain triples $(Qx_i, Kx_i, Vx_i)$ for each $i \in \{1, \ldots, m\}$, called query vectors, key vectors, and value vectors. We abbreviate these as $(q_i, k_i, v_i)$. Next, for each $i, j$, dot-product attention computes logits using dot products $q_i \cdot k_j$. These logits are in turn used to compute an output matrix $[\text{softmax}(q_i \cdot k_1 / \sqrt{m}, \ldots, q_i \cdot k_m / \sqrt{m}) \cdot v_j]_{i,j}$. We denote this output matrix as $\text{SA}(x_1, \ldots, x_m)$. Cross-attention is similar, accepts inputs that are partitioned into disjoint blocks $(y_1, \ldots, y_n)$ and $(x_1, \ldots, x_m)$, and only applies $Q$ to $y$ and only applied $K, V$ to $x$. We refer to this as $\text{CA}(y_1, \ldots, y_n, x_1, \ldots, x_m)$.

**Attention for Input-Output Relationships.** Exploiting semantic relationships between inputs and outputs via an attention-based model has been studied in the deep learning literature. In natural language processing, such an idea is commonly used in question answering models (dos Santos et al., 2016; Tan et al., 2016; Yang et al., 2016). For instance, Yang et al. (2016) form a matrix that represents the semantic matching information of term pairs from a question and answer pair, and then use dot-product attention to model question term importance. For regression tasks, Kim et al. (2019) proposed attentive neural processes (ANP) that use dot-product attention to allow each input location to attend to the relevant context points for the prediction, and applied ANP to vision problems.

**Human Coordination.** Our work is also inspired by how humans coordinate in cooperative settings. Theory-of-mind, the mechanism people use to infer intentions from the actions of others, plays a key role in structuring coordination (Wu et al., 2021; Shum et al., 2019). In particular, rational speech acts (RSA) is an influential model of pragmatic implicature (Frank & Goodman, 2012; Goodman & Stuhlmüller, 2013). At the heart of these approaches are probabilistic representations of beliefs that allow for modeling uncertainty and recursive reasoning about the beliefs of others, enabling higher-order mental state inferences. This recursive reasoning step also underlies the cognitive hierarchy and level-K reasoning models, and is useful for explaining certain focal points (Camerer, 2011; Stahl & Wilson, 1995; Camerer et al., 2004). However, constructing recursive models of players' beliefs and behaviors is computationally expensive as each agent must construct an exponentially growing number of models of each agent modeling each other agent. As a result, recursive models are often limited to one or two levels of recursion. Furthermore, none of these approaches can by itself take advantage of the shared features across actions and observations.

## 3 DEC-POMDPS WITH SHARED ACTION AND OBSERVATION FEATURES

It is common to describe the states and observations in Dec-POMDPs using features, e.g. in card games each card has a rank and a suit. These featurized observations can be exploited by function approximators. In contrast, in typical MARL implementations the actions are merely outputs of the neural network and the models do not take advantage of features of the actions. In the standard representation of Dec-POMDPs, actions are defined solely through their effect on the environment through the reward and the state transition functions. In contrast, in real world environments actions are often grounded and can be described with semantic features that refer to the object they act on, e.g. "I pull the *red lever*".

To allow action features to be used by MARL agents, we formalize the concept of observation and action features in Dec-POMDPs. We say a Dec-POMDP has *observation features* if, for at least one player $i$, we can represent the observation $\Omega^i(s)$ as a set of $\ell$ objects $\Omega^i(s) = \{O_1, \ldots, O_\ell\}$, where each object $O_j = (f_1, \ldots, f_{n_j})$ is described by a vector of $n_j$ features. Each of these features $f_k$ exists in a feature space $F_k$. Similarly, a Dec-POMDP has *action features* if one can factor the representation of the actions into features $a^i = (\hat{f}_1, \ldots, \hat{f}_m)$, where each action feature $\hat{f}_r \in \hat{F}_r$, $r = 1, ..., m$, and $\hat{F}_r$ is the action feature space.

In some Dec-POMDPs, actions can be described using some of the *same* features that describe the observations. For example, an agent might observe the "red" light and take the action of pulling the "red" lever where "red" is a shared feature between observations and actions. In such cases there is a *non-empty intersection* between $F_k$ and $\hat{F}_r$ ("shared action-observation features") which may be exploited for coordination. Even in the absence of an exact match, the distance between similar features (e.g., "pink" and "red" and vs. "green" and "red") might also be useful for coordination.

## 4  THE HINT-GUESS GAME

To study Dec-POMDPs with shared action-observation features, we introduce a novel setting that we call *hint-guess*. Hint-guess is a two-player game where players must coordinate to successfully guess a target card. The game consists of a *hinter* and a *guesser*. Both players are given a hand of $N$ cards, $H_1 = \{C_1^1, ...C_N^1\}$ for the *hinter* and $H_2 = \{C_1^2, ...C_N^2\}$ for the *guesser*. Each card has two features $(f_1, f_2)$ where $f_1 \in F_1$ and $f_2 \in F_2$. Cards in each hand are drawn independently and randomly with replacement, with equal probability for each combination of features. Both hands, $H_1$ and $H_2$, are public information exposed to both players. Before each game, one of the *guesser's* cards, $C_i^2$, is randomly chosen to be the target card and its features are revealed to the *hinter*, but not the *guesser*.

In the first round, the *hinter* (who observes $H_1, H_2, C_i^2$) chooses a card in its own hand, which we refer to as $C_j^1$, to show to the *guesser*. In the second round, the *guesser* (who observes $H_1, H_2, C_j^1$) guesses which of its cards is the target. Both players receive a common reward $r = 1$ if the features of the card played match those of the target, otherwise $r = 0$ for both players.

Fig. 2 shows some simple scenarios that probe key dimensions of coordination with $N = 2$, $F_1 = \{1, 2, 3\}$ and $F_2 = \{A, B, C\}$. Each of these scenarios has a human-compatible and intuitive solution. The first scenario (exact match) is the most simple—the *hinter* has a copy of the target card (2B) so it can simply hint 2B. The next scenario (feature similarity) requires reasoning about the features under some ambiguity since neither of the cards in the two hands are a direct match. In this case, both cards in the *hinter*'s hand share one feature with the *guesser*. Thus, the human-compatible strategy would be to match the cards that share features to each other. The third and fourth examples (labeled implicatures in Fig. 2) require understanding the action embedded within its context, e.g. what the *hinter* would have done had the goal been different. The third scenario invokes a simple kind of implicature: mutual exclusivity. In this scenario, human-compatible intuitive reasoning follows the logic of: "if the target card *was* 1B, the *hinter* would choose 1B. So that means 1B is taken and 3C should correspond to 2A even though they share no common feature overlap". The final scenario combines feature similarity and mutual exclusivity. These scenarios are particularly interesting as deep learning models often struggle to effectively grapple with mutual exclusivity (Gandhi & Lake, 2020).

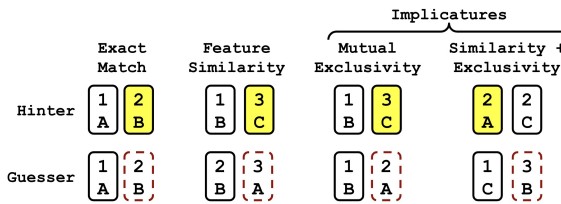

Figure 2: Example scenarios in *hint-guess*. Shown above are four hand-crafted scenarios that test distinct dimensions important for ZSC. The card in the red dotted lines is a target card and the highlighted yellow card corresponds to a human-compatible choice. The two right scenarios require agents to reason about implicatures, i.e., the intuitive choice has zero feature overlap with the target card. Model performance in these scenario types is shown in Table 2.

## 5  ARCHITECTURES EXAMINED

We consider the following architectures to investigate the effect of policy parameterization on the agents' ability to learn intuitive relationships between actions and observation. For details about the model architectures, see Appendix A.1.

**Feedforward Networks (MLPs).** The most basic architecture we test is a standard fully connected feedforward network with ReLU activations. All featurized representation of objects in the observation are concatenated and fed into the network, which outputs the estimated Q-value for each

action. There is no explicit representation of action-observation relationships in this model, since observations are inputs and actions are outputs.

**Feedforward Networks with Action as Input (MLP Action In)** We also examined variant of the MLP architectures in which featurized representation of objects in the observation *and one particular action* are concatenated and fed into the network, which outputs the estimated Q-value for the action that was fed in. Note that this requires a separate forward pass to compute the Q-value for each action.

**Attention (Attn).** We also investigate three attention-based models as shown in Fig. 3. The first model processes the observations using attention, takes the object-wise mean, and feeds the output into a feedforward network, which produces a vector with a Q-value for each action

$$Q = \text{MLP}(\text{Mean}(\text{SA}(O_1, \ldots, O_n))).$$

There is no explicit representation of action-observation relationships in this model, since observations are inputs and actions are outputs.

**Cross Attention with Action as Input (CA2I).** In the second attention model, the featurized actions are fed into a cross-attention block as queries and the featurized observations are fed in as keys and values, which produces a vector with a Q-value for each action

$$Q = \text{MLP}(\text{Mean}(\text{CA}(A_1, \ldots, A_m, O_1, \ldots, O_n))).$$

**Self Attention with Action as Input (SA2I).** Lastly, we look at an attention-based architecture similar to Attn, where a featurized action is passed as input to the attention module(s) along with the observations. This outputs a single scalar value at a time, the estimated Q-value for the specific action being fed into the network

$$Q_k = \text{MLP}(\text{Mean}(\text{SA}(O_1, \ldots, O_n, A_k))).$$

for $k = 1, \ldots, m$. Indeed, SA2I requires a forward pass for each action to calculate the Q-value vector.

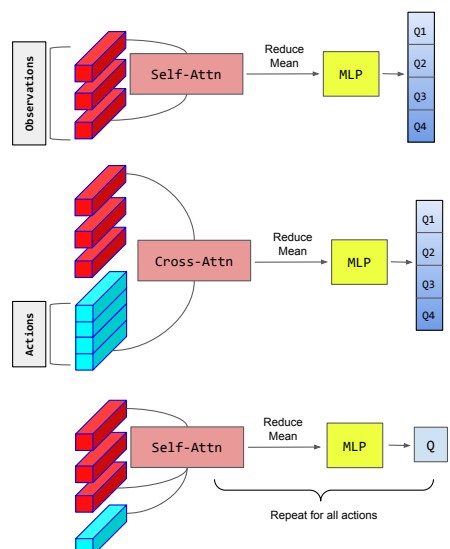

Figure 3: Model architecture for the attention-based models. Top: Attention (Attn). Middle: Cross Attention with Action as Input (CA2I). Bottom: Self Attention with Action as Input (SA2I). The red blocks denote featurized objects in the observation, e.g. cards in the deck. The cyan blocks denote featurized actions, e.g. cards in the hand that can be hinted/guessed. *Cross-Attn*, *Self-Attn* and *MLP* denote the cross attention, self attention, and fully-connected layers, respectively.

Note that, of all the architectures examined here, only SA2I processes the featurized actions and observations using the same weights. Indeed, MLP and Attn do not consider action features at all; MLP Action In uses a different vector of weights for each input dimension; and CA2I uses a different set of weights for the queries (i.e., the actions) than it uses for the keys and values (i.e., the observations). We hypothesize that this contrast may explain the strong performance of SA2I that we will describe in the following sections.

# 6 EXPERIMENTS

## 6.1 EXPERIMENT SETUP

We experimentally evaluate the architectures in the hint-guess game introduced in Section 4. In Sections 6.2-6.4, we fix the hand size to be $N = 5$ and the features to be $F_1 = \{1, 2, 3\}$ and $F_2 = \{A, B, C\}$. [1] We use a one-hot encoding for features; more specifically, we use a two-hot vector to represent the two features of each card. In Sec. 6.5, we examine a qualitatively different version of the game where $N = 3$ and there is only one feature, $F_1 = \{0, 1, ..., 19\}$. In this version,

---

[1]There is nothing particular about the hand size, and as shown in Appendix A.5, similar results can be obtained with either a larger or smaller hand size.

we investigate whether it is possible to capture ordinal relationships between actions using sinusoidal positional encodings. For these experiments, we encode each number as a 200-dimensional vector consisting of sine and cosine functions of different frequencies, as in Vaswani et al. (2017).

For both variants of the game, the observation input is a sequence of card representations for both hands $H_1$ and $H_2$, as well as the representation of the target card, $C_i^2$ (for the *hinter*) or the hinted card $C_j^1$ (for the *guesser*); a binary feature is added to each card's featurization specifying to which player it belongs. We train agents in the standard self-play setting using independent Q-learning (Tan, 1993, IQL), where the *hinter* and *guesser* are jointly trained to maximize their score in randomly initialized games; the hinter and guesser do not share any weights. To avoid giving the set-based attention architectures an unfair advantage, we also permute the cards in the hands observed by all agents so that agents are not able to coordinate using the position of the cards. The action space of the architectures that output multiple Q-values is of the form "hint $(f_1, f_2)$" for each $(f_1, f_2)$; we mask the Q-values so that only features corresponding to cards in the player's hand can be chosen. To evaluate success, we consider the agents' performance and behavior in both SP and the intra-AXP setting. We also provide fine-grained examination of their policies and investigate their ability to match the human-compatible response in different scenarios. See Appendix A.1 for training details.

## 6.2 CROSS-PLAY PERFORMANCE.

First, we evaluate model cross-play (XP) performance for each architecture in the intra-AXP setting. In this setting, agents from independent training runs with different random seeds are paired together. Fig. 4 records the scores obtained by each pair of agents, where the diagonal entries are the within-pair SP scores and the off-diagonal entries are XP scores. Table 1 summarizes average SP and XP scores across agents.

**Comparison Across Architectures.** Fig. 4 shows that the XP matrix of all architectures except SA2I (Self Attention with Action as Input) lack an interpretable pattern. The XP score is near chance for these architectures as shown in Table 1. In contrast, the XP matrix for the SA2I model shows two clear clusters. Within the clusters, agents show XP performance nearly identical to that of their SP, implying that they coordinate nearly perfectly with other agents trained with a different seed, whereas outside the clusters they achieve a return close to zero. As we will show in the next section, the upper cluster, which has a higher average XP score, corresponds to a highly interpretable and human-like strategy where agents *maximize* the "similarity" between the target and the hint card (as well as between the hint card and the guess card). In the lower, second cluster, agents do the opposite. They try to hint/guess cards that share no common feature with the target/hint cards. In the rest of the paper, we will refer to the cluster where agents maximize the similarity between cards as **SA2I Sim**, and the cluster where agents maximize the dissimilarity as **SA2I Dissim**. However, as we will see in section 6.3, the SA2I agents do not just maximize/minimize feature similarity; they also demonstrate more sophisticated coordination patterns that exploit implicature.

| Model Architectures | | |
|---|---|---|
| Model | Cross-Play | Self-Play |
| MLP | $0.27 \pm 0.04$ | $0.85 \pm 0.02$ |
| MLP Action In | $0.28 \pm 0.05$ | $0.87 \pm 0.02$ |
| Attn | $0.27 \pm 0.04$ | $0.87 \pm 0.01$ |
| CA2I | $0.29 \pm 0.03$ | $0.85 \pm 0.02$ |
| SA2I | $0.37 \pm 0.12$ | $0.76 \pm 0.02$ |
| SA2I Sim | $0.77 \pm 0.01$ | $0.82 \pm 0.01$ |
| SA2I Dissim | $0.71 \pm 0.01$ | $0.72 \pm 0.01$ |

| Baseline Training Algorithms | | |
|---|---|---|
| Algothrim | Cross-Play | Self-Play |
| OP | $0.35 \pm 0.02$ | $0.35 \pm 0.02$ |
| OBL (level 1) | $0.27 \pm 0.05$ | $0.29 \pm 0.06$ |
| OBL (level 2) | $0.28 \pm 0.04$ | $0.28 \pm 0.05$ |

Table 1: Cross-play performance in the intra-AXP setting. Each entry is the average performance of 20 pairs of agents that are trained with different random seeds. The XP score is the off-diagonal mean of each grid. The SP score is the diagonal mean, i.e. the score attained when agents play with the peer they are trained with.
Note that a "chance agent" that acts randomly is expected to obtain a score of 0.28 in this setting.
All models in the "Model Architecture" part are trained with IQL (Tan, 1993), and all algorithms in the "Baseline Training Algorithm" section use an MLP architecture.

**Comparison with intra-AXP Baselines.** The bottom part of Table 1 contains the SP and XP results for two recent intra-AXP algorithms, other-play (Hu et al., 2020, OP) and off-belief learning (Hu et al., 2021, OBL). For details and implementation of the baseline algorthims, see Appendix A.3.

As shown, the XP scores for OP agents only show marginal improvement over MLP agents. By preventing arbitrary symmetry breaking, OP improves XP performance, but only to a limited extent. In contrast, the OBL agents fail to obtain scores beyond chance both in XP and SP. This is expected, as OBL is designed to explicitly prevent the interpretation of cheap *cheap talk*, i.e., costless messages between players, which is important for coordination in hint-guess.

## 6.3 POLICY EXAMINATION

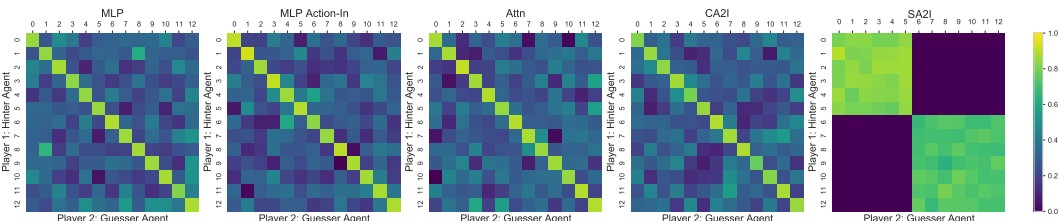

Figure 4: Cross-play matrices. Cluserting visualization of paired evaluation of different agents trained under the same method. The y-axis represents the agent index of the *hinter* and the x-axis represents the agent index of the *guesser*. Each block in the grid is obtained by evaluating the pair of agents on 10K games with different random seeds. Numerical performance is shown in Table 1.

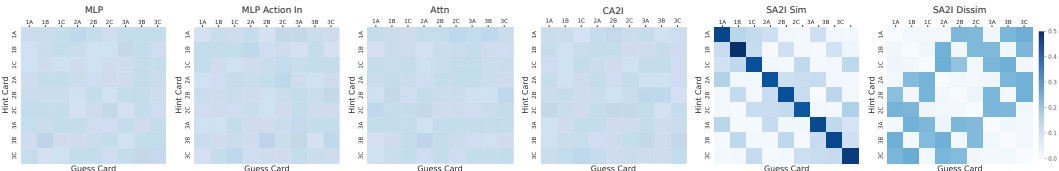

Figure 5: Conditional probability matrices. We show $\Pr$ (Guess|Hint), for the *guesser* to guess a particular card (x-axis) when the hinted card is the card on the y-axis. Each subplot is the sample average of 20 agent-pairs with different seeds for 1K games within pair. $\Pr$ (Hint|Target) look almost identical so is omitted.

**Conditional Probability Analysis.** In Fig. 5, we provide the conditional probability for the *guesser* to guess a card given the hinted card (bottom row). One crucial thing to analyze is whether agents assign different probabilities to actions based on the features they share with the observation. One can see that for MLP, MLP Action In, Attn, and CA2I the probability matrices for both target-hint and hint-guess are nearly uniform. This implies that the SP policies across seeds each form their own private language for arbitrary and undecipherable coordination.

In contrast, for the two clusters of SA2I agents, the correlation (or anti-correlation) between the action features and target/hint card features is much stronger. For SA2I Sim, both the *hinter* and the *guesser* prioritize exact matches when they are present. If the exact match is not present, they turn to cards that share one feature in common. The SA2I Dissim agents do the exact opposite—matching cards together that share as few features as possible.

**Human Compatibility Analysis.** However, we find that the nuance with which these clusters play goes beyond simply maximizing or minimizing feature similarity. To demonstrate this, we run simulations on the four scenarios (exact match, feature similarity, mutual exclusivity, exclusivity+similarity) shown in Fig. 2 and described in Section 4. In Table 2, we record the percentage of times where SA2I agents in each cluster chose the human-compatible actions in Fig. 2. We find that SA2I Sim agents demonstrate coordination patterns that are nearly identical to a human-compatible policy. These results are surprising given that our models have never been trained with any human data. Furthermore, mutual exclusivity was thought to be hard for deep learning models to learn (Gandhi & Lake, 2020). In contrast, while the SA2I Dissim agents always perform actions that are the *opposite* to the human-compatible policy, these conventions are still interpretable and non-arbitrary.

## 6.4 HUMAN-AI EXPERIMENTS

We recruited 10 university students to play hint-guess. Each subject played as *hinter* for 15 randomly generated games, totaling 150 different games. These subjects are then cross-matched to play as

| Scenario | Self-Play (SA2I Sim) | | Cross-Play (SA2I Sim) | | Self-Play (SA2I Dissim) | | Cross-Play (SA2I Dissim) | |
|---|---|---|---|---|---|---|---|---|
| | Human (%) | Win (%) | Human (%) | Win (%) | Human (%) | Win (%) | Human (%) | Win (%) |
| Exact match | 100.0 | 100.0 | 100.0 | 100.0 | 0.0 | 100.0 | 0.0 | 100.0 |
| Feature similarity | 100.0 | 100.0 | 100.0 | 100.0 | 0.0 | 100.0 | 0.0 | 100.0 |
| Mutual exclusivity | 100.0 | 100.0 | 100.0 | 100.0 | 9.3 | 91.2 | 9.3 | 92.2 |
| Similarity + Exclusivity | 92.0 | 91.7 | 97.9 | 99.5 | 3.2 | 98.4 | 0.0 | 99.9 |

Table 2: Behavioral analysis for the SA2I model in the Fig. 2 scenarios. We randomly chose 20 agent-pairs from each cluster and simulated the same scenario 1K times. Human (%) denotes the fraction of games where the *hinter* hints the card that corresponds to human-compatible choice (highlighted in yellow in Fig. 2), and Win (%) denotes the fraction where the *guesser* correctly guesses.

*guessers* with the hints their peers generated. The human hints are also fed into randomly sampled MLP and SA2I Sim *guesser*-agents to test AI performance against human partners. The experiment was carefully designed so that the hinter is never informed of the guesser's guess and the guesser is never informed of the true target card. This experimental design ensures that the human participants generate zero-shot data, and do not optimize their play using previous experience. Further details of the experiment are in Appendix A.2.

**Ad-Hoc Performance.** In the right table of Fig. 6 we report average ad-hoc coordination scores obtained by *hinter-guesser* pairs for human-human, human-MLP, and human-SA2I Sim. Humans obtained an average ad-hoc coordination score of 0.75 with their peers. As a baseline, the MLP *guessers* show poor performance in understanding human-generated hints, barely outperforming random guessing. In contrast, the SA2I Sim *guessers* achieve human-level performance with an average ZSC score of 0.77 with humans. Note that this score is very close to the average ZSC score in Table 1, where SA2I Sim agents cross-played among themselves.

**Human-AI Behavior Correlation.** We also investigate two kinds of correlations between human play and AI play. The right table of Fig. 6 shows the percentage of games where model *guessers* chose the same action as the human *guessers*. In 80.7% of the games, the SA2I Sim agents and human *guessers* agree on the same action across many different scenarios. In contrast, the MLP *guessers* deviate from human *guessers*, with only 40.7% agreement.

**Human-AI Performance Correlation.**
The left plot of Fig. 6, shows the correlations between human-human play and human-AI play. Across humans we observed a range of skill levels at the game, with some hinters not even achieving 50% guess accuracy when paired with other humans, while others exceeded 90% (as measured along the x-axis). We see the performance of human-SA2I Sim pairs increased substantially with the skill level of the human, whereas the performance of human-MLP pairs was less sensitive along this axis. Taken together, these results suggest that the SA2I Sim agent is both better at coordinating with people than a baseline model and is also

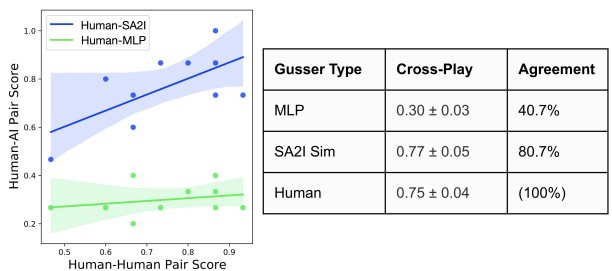

Figure 6: Human-AI ZSC results comparing human-human pairs, human-SA2I Sim pairs and human-MLP pairs. In the left plot, each point corresponds to a particular human hinter. In the table, "agree" measures the percentage of games in which the guesser selected the same card as the human guesser.

better at coordinating with the people who are better at coordinating with people.

## 6.5 SINUSOIDAL ENCODING, MULTI-HEAD AND MULTI-LAYER ATTENTION

In previous settings, the SA2I models were able to learn sophisticated cognitive patterns like mutual exclusivity from one-hot encoding of inputs (equal or non-equal). However, one-hot encoding does not capture richer semantic relationships between features. For instance, in hint-guess, if cards are encoded one-hot, the agent can only "know" that the card 1A has the same first label as the card 1B, but it cannot "know" whether the number 1 is closer to 2 than to 5. Thus, in this section, we investigate whether a more expressive encoding enables the SA2I model to learn to leverage

the ordinal relationship between features. Specifically, we examine the performance of sinusoidal positional encodings in a variant of hint-guess in which the only card feature is a number between 0 and 19, as described in the experiment setup.

Table 3 shows SP and XP performance of SA2I agents with one-hot encoding and sinusoidal encoding in this single-feature setting. Agents with one-hot encoding are near chance in XP. They do not form clusters as observed before. We hypothesize that the failure of one-hot agents is because of the large feature space (20 numbers) relative to the small number of features (1). Specifically, since the feature space is large and the agents are only sensitive to exact overlap, they degenerate into using arbitrary conventions, resulting in a large performance gap between SP and XP.

Agents with sinusoidal encodings, in contrast, split into two clusters (named SA2I Sim and SA2I Dissim as before), wherein each cluster has near-perfect SP and XP scores with no significant performance gap. We find that these agents learn to exploit the ordering and distance information between the numbers for coordination. SA2I Sim agents rank the *hinter*'s and *guesser*'s hands in the same order and match the corresponding numbers as hint-guess pairs. SA2I Dissim agents, on the other hand, rank one hand in ascending order and the other hand in descending order for matching. See Fig. 7 for a concrete example. For both strategies, if the hinter does not have duplicate numbers in its hand, agents obtain a near-perfect play score. Indeed, in a XP simulation across 15

| Encoding | One-hot | Sinusoidal |
|---|---|---|
| SP | 0.81 ± 0.02 | 0.92 ± 0.01 |
| XP | 0.36 ± 0.10 | 0.52 ± 0.16 |
| XP (SA2I Sim) | - | 0.92 ± 0.01 |
| XP (SA2I Dissim) | - | 0.93 ± 0.01 |

Table 3: SP and XP scores for SA2I agents with one-hot and sinusoidal encodings. Agents with sinusoidal encoding form two clusters so we also show the within-cluster results.

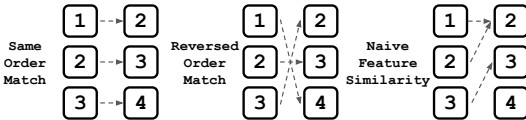

Figure 7: A scenario illustrating the behavior of SA2I agents. In this scenario, the *hinter*'s hand is $(1, 2, 3)$ and the *guesser*'s hand is $(2, 3, 4)$ (the actual hands seen by agents will be permuted). We find that with probability close to 1, SA2I Sim agents use a strategy that exploits *same order matching* (left). They sort both hands in the same order and match 1-2, 2-3, 3-4, etc. Also with probability close to 1, SA2I Dissim agents use *reversed order matching* (middle). They sort one hand in ascending order and the other in descending order and match. To compare we also show *naive feature similarity* (right) that solely maximizes feature similarity; this strategy will match 1-2, 2-2, 3-3 and leave out 4.

agent-pairs with 1K games per pair, where each agent's hand is drawn *without* replacement (so no duplicates), in 99.9% of the time, the SA2I Sim agents hint/guess exactly according to the same order matching scheme. Also in 99.9% of the time, the SA2I Dissim agents hint/guess according to the reversed order matching scheme.

We also find that the results for SA2I architectures are qualitatively similar when using multi-head or multi-layer attention or both in Appendix A.4. This suggests that SA2I may also be able to produce human-compatible policies in settings where larger architectures are required for effective learning.

# 7 CONCLUSION AND FUTURE WORK

We investigated the effect of network architecture on the ability of learning algorithms to exploit the semantic relationship between shared features across actions and observations for coordination. We found that self-attention-based architectures which jointly process a featurized representation of observations and actions have a better inductive bias for exploiting this relationship; we hypothesize that this difference in performance may be explained by whether the architectures use the same weights to process both the action features and the observation features.

For future work, we think it would be interesting to investigate the behavior of the SA2I architecture in settings with more abstract features (compared to the categorical and ordinal features investigated in this work), like word embeddings or multi-modal features. Another interesting future direction would be to use SA2I on a sequence of actions to capture the abstract correspondence between actions across different time steps. One limitation of this work is that action features need to be provided to the agent, which is a stronger assumption than is typical for Dec-POMDPs. In some situations, it may be possible to learn action features, and this is another direction for future work.

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

# A APPENDIX

## A.1 TRAINING AND MODEL ARCHITECTURE DETAILS

**Training Setup.** By default we use standard experience-replay with a replay memory of size 300K. For optimization, we use the mean squared error loss, stochastic gradient descent with learning rate set to $10^{-4}$ and minibatches of size 500 each. We train the agents using 4M episodes. To allow more data to be collected between training steps, we update the network only after we receive 500 new observations rather than after every observation. We use the standard exponential decay scheme with exploration rate $\epsilon = \epsilon_m + (\epsilon_0 - \epsilon_m) \exp(-n/K)$, where $n$ is the number of episodes, $\epsilon_m = 0.01$, $\epsilon_0 = 0.95$, and $K = 50,000$. All experiments were run on two computing nodes with 256GB of memory and a 28-Core Intel 2.4GHz CPU. A single training run takes roughly 8 hours for the SA2I model and 2 hours for all other models. All *hinters* and *guessers* have the same model structure.

The model architecture details are as follows:

**Feedforward Networks (MLP).** The MLP agent is a feedforward ReLU neural network with 3 hidden layers (width 128).

**Attention Models.** All three attention-based models share the same single-head attention layer with size determined by the shapes of the input and output. By default we do not add position encoding. In Attn and SA2I, we add a feedforward ReLU MLP with 3 hidden layers (width 128 each) after the attention layer. In CA2I, we add fully-connected layers with conformable dimensions but do not add extra ReLU activation.

## A.2    DESIGN OF HUMAN-AI EXPERIMENTS

In this section we provide details of the design of human-based experiments.

We recruited 10 individuals who are undergraduate and master students at a university. The instructions they received include: (i) rules of hint guess; (ii) that they need to assume they are playing against a *guesser* who is an ordinary human (they are not told that they would play against AI bots); (iii) that the position of the cards are permuted so they cannot provide/interpret hints based on card position, and (iv) if two cards have the same features they are effectively the same.

After they showed understanding of the instructions, the subjects were asked to act as *hinters*. Each subject was provided with 15 randomly generated games with $F_1 = \{1, 2, 3\}$, $F_2 = \{A, B, C\}$ and $N = 5$. They were presented with both hands and the target card and were asked to write down their hints. A sample question they received was:

```
            Playable card is:  2B
  Your (hinter) hand is:  1C, 1B, 3B, 2C, 1A
      Guesser hand is:  1B, 2C, 2B, 2C, 2B
Please choose a hint card from YOUR OWN hand!
```

After the subjects provided hints, they were given no feedback to ensure that they could not learn about the agent they are playing with; this ensures that the setting remains ad hoc.

Then we used the hints provided by the subjects to obtain human-human (*hinter-guesser*) and human-AI scores. To obtain human/human scores, we randomly mix-matched the subjects so that each subject would now act as the *guesser* for 15 games generated by another human subject. This time, they were provided with both hands and the human hint, and were ask to choose one card to play. Same as before, they did not receive any feedback about their guesses to ensure that the setting remained ad hoc.

To obtain human-SA2I Sim and human- MLP scores, we reproduced the 150 games and fed human hints to randomly sampled SA2I Sim-MLP *guesser* agents.

### A.3  DETAILS OF INTRA-AXP BASELINES

In this section we provide implementation details of other-play (Hu et al., 2020) and off-belief learning (Hu et al., 2021), two recent intra-algorithm cross play (intra-AXP) methods that we use as baselines.

**Other-play (OP).** The goal of OP is to find a strategy that is robust to partners breaking symmetries in different ways. To achieve this, it uses reinforcement learning to maximize returns when each agent is matched with agents playing the same policy, but under a random relabeling of states and actions under known symmetries of the Dec-POMDP (Hu et al., 2020). To apply OP, we change the objective function of the hinter from the standard self-play (SP) learning rule objective

$$\pi^* = \arg\max_\pi J\left(\pi^1, \pi^2\right) \tag{1}$$

To the OP objective

$$\pi^* = \arg\max_\pi \mathbb{E}_{\phi \sim \Phi} J\left(\pi^1, \phi\left(\pi^2\right)\right) \tag{2}$$

where the expectation is taken with respect to a uniform distribution on $\Phi$, where $\Phi$ describes the symmetries in the underlying Dec-POMDP (in the hint-and-play games, the symmetries are the two features).

As OP only changes the objective function, it can be applied on top of any SP algorithm. We choose to apply OP on top of the MLP architecture, using the same training method as detailed in Appendix. A.1, and change the objective to the OP objective. In implementation, this means that the guesser will receive a feature-permuted version of the game, i.e. feature 1 and feature 2 (the letters and the numbers) of the hands and the hint that the guesser receives will be a permuted version of what the hinter originally receives.

**Off-belief learning (OBL).** OBL (Hu et al., 2021) regularizes agents' ability to make inferences based on the behavior of others by forcing the agents to optimize their policy $\pi_1$ assuming past actions were taken by a given fixed policy $\pi_0$, while in the same time assuming that future actions will be taken by $\pi_1$. In practice, OBL can be iterated in a hierarchical order, where the optimal policy from the lower level becomes the input to the next higher level.

We apply OBL on MLP agents and keep the training setup the same as in Appendix A.1. At the lowest level (level 1), OBL agents assume $\pi_0$ is the policies where actions are chosen uniformly at random. And OBL level 2 assumes the policy from OBL level 1 is the new $\pi_0$, and so forth.

## A.4 MULTI-HEAD AND MULTI-LAYER ATTENTION

We show the XP results when using the SA2I architecture with different number of attention heads or attention layers. We also show the same results for using either one-hot or sinusoidal encoding. To enable using sinusoidal encoding, we fix the hand size $N = 3$ and only use one feature, $F_1 = \{0, 1, ..., 19\}$, which is equivalent to fixing $F_2 = \{A\}$.

For these experiments we use standard experience-replay with a replay memory of size 1K. For optimization, we use the mean squared error loss, stochastic gradient descent with learning rate set to $10^{-4}$ and minibatches of size 200 each. We train the agents using 5M episodes. To allow more data to be collected between training steps, we update the network only after we receive 50 new observations rather than after every observation. We use the standard exponential decay scheme with exploration rate $\epsilon = \epsilon_m + (\epsilon_0 - \epsilon_m) \exp(-n/K)$, where $n$ is the number of episodes, $\epsilon_m = 0.05$, $\epsilon_0 = 0.95$, and $K = 15,000$. All experiments were run on two computing nodes with 256GB of memory and a 28-Core Intel 2.4GHz CPU. A single training run takes roughly 6 hours for 1-layer attention and 18 hours for 3-layer attention. All *hinters* and *guessers* have the same model structure.

| Encoding | Layers | Heads | SP | XP | XP (Clus. 1) | XP (Clus. 2) |
|---|---|---|---|---|---|---|
| One-hot | 1 | 1 | $0.81 \pm 0.02$ | $0.36 \pm 0.10$ | - | - |
| One-hot | 1 | 4 | $0.90 \pm 0.02$ | $0.36 \pm 0.10$ | - | - |
| One-hot | 3 | 1 | $0.40 \pm 0.04$ | $0.38 \pm 0.14$ | - | - |
| One-hot | 3 | 4 | $0.89 \pm 0.05$ | $0.36 \pm 0.11$ | - | - |
| Sinusoidal | 1 | 1 | $0.92 \pm 0.01$ | $0.52 \pm 0.16$ | $0.92 \pm 0.01$ | $0.93 \pm 0.01$ |
| Sinusoidal | 1 | 4 | $0.92 \pm 0.01$ | $0.67 \pm 0.13$ | $0.92 \pm 0.01$ | $0.93 \pm 0.01$ |
| Sinusoidal | 3 | 1 | $0.94 \pm 0.01$ | $0.56 \pm 0.16$ | $0.95 \pm 0.01$ | $0.91 \pm 0.01$ |
| Sinusoidal | 3 | 4 | $0.96 \pm 0.01$ | $0.76 \pm 0.09$ | $0.95 \pm 0.01$ | $0.95 \pm 0.01$ |

Table 4: Robustness results for using multi-head and multi-layer attention. We show SP and XP scores for SA2I agents with different number of attention modules (layers), attention heads, and different encoding of features. Each entry is the average performance is of 20 pairs of agents that are trained with different random seeds. Agents with sinusoidal encoding forms two clusters so we also show the within-cluster results.

A.5 EFFECT OF HAND SIZE ON PERFORMANCE

In this section we analyze the effect of hand size on agents' self-play and cross-play performance. We fix the features to be $F_1 = \{1, 2, 3\}$ and $F_2 = \{A, B, C\}$ and run experiments with hand size $N = \{3, 7\}$, as opposed to $N = 5$ in the main text. As shown in the following Table 5, SP and XP results largely confirm our findings for $N = 5$. We also show the same results for a simplified version of the game, where both *hinter* and *guesser* have the same hand. In the simplified game, an optimal and intuitive strategy exists, which is to always hint the target card and then guess the card that is hinted. The training setup and model details are exactly the same as in A.1.

| Method | Full Game ($N = 3$) | | Same Hand Condition | |
| --- | --- | --- | --- | --- |
| | Cross-Play | Self-Play | Cross-Play | Self-Play |
| MLP | $0.47 \pm 0.06$ | $0.92 \pm 0.01$ | $0.87 \pm 0.04$ | $0.98 \pm 0.02$ |
| MLP Action In | $0.47 \pm 0.07$ | $0.92 \pm 0.01$ | $0.70 \pm 0.04$ | $0.99 \pm 0.01$ |
| Attn | $0.47 \pm 0.06$ | $0.92 \pm 0.01$ | $0.47 \pm 0.05$ | $0.95 \pm 0.01$ |
| CA2I | $0.46 \pm 0.05$ | $0.81 \pm 0.01$ | $0.46 \pm 0.06$ | $0.88 \pm 0.04$ |
| SA2I | $0.43 \pm 0.12$ | $0.80 \pm 0.01$ | $0.45 \pm 0.18$ | $0.95 \pm 0.02$ |
| SA2I Sim | $0.81 \pm 0.01$ | $0.81 \pm 0.01$ | $1.00 \pm 0.00$ | $1.00 \pm 0.00$ |
| SA2I Dissim | $0.80 \pm 0.01$ | $0.80 \pm 0.01$ | $0.89 \pm 0.02$ | $0.90 \pm 0.01$ |
| OP | $0.55 \pm 0.03$ | $0.54 \pm 0.04$ | $0.90 \pm 0.01$ | $0.98 \pm 0.01$ |
| OBL (level 1) | $0.47 \pm 0.07$ | $0.44 \pm 0.06$ | $0.47 \pm 0.03$ | $0.48 \pm 0.03$ |
| OBL (level 2) | $0.47 \pm 0.04$ | $0.46 \pm 0.06$ | $0.47 \pm 0.05$ | $0.47 \pm 0.05$ |

| Method | Full Game ($N = 7$) | | Same Hand Condition | |
| --- | --- | --- | --- | --- |
| | Cross-Play | Self-Play | Cross-Play | Self-Play |
| MLP | $0.22 \pm 0.08$ | $0.72 \pm 0.03$ | $0.77 \pm 0.10$ | $0.96 \pm 0.08$ |
| MLP Action In | $0.25 \pm 0.09$ | $0.76 \pm 0.01$ | $0.62 \pm 0.14$ | $0.95 \pm 0.05$ |
| Attn | $0.24 \pm 0.09$ | $0.77 \pm 0.03$ | $0.24 \pm 0.08$ | $0.93 \pm 0.01$ |
| CA2I | $0.22 \pm 0.11$ | $0.77 \pm 0.02$ | $0.22 \pm 0.13$ | $0.80 \pm 0.09$ |
| SA2I | $0.37 \pm 0.33$ | $0.70 \pm 0.06$ | $0.40 \pm 0.38$ | $0.89 \pm 0.11$ |
| SA2I Sim | $0.78 \pm 0.01$ | $0.78 \pm 0.03$ | $1.00 \pm 0.00$ | $1.00 \pm 0.00$ |
| SA2I Dissim | $0.67 \pm 0.02$ | $0.67 \pm 0.02$ | $0.67 \pm 0.03$ | $0.67 \pm 0.02$ |
| OP | $0.32 \pm 0.02$ | $0.31 \pm 0.02$ | $0.79 \pm 0.02$ | $0.95 \pm 0.02$ |
| OBL (level 1) | $0.21 \pm 0.04$ | $0.22 \pm 0.03$ | $0.22 \pm 0.03$ | $0.22 \pm 0.04$ |
| OBL (level 2) | $0.24 \pm 0.03$ | $0.22 \pm 0.03$ | $0.22 \pm 0.03$ | $0.25 \pm 0.02$ |

Table 5: Cross-play performance for card number $N = 3$ and $N = 7$. We show results for both the full hint guess game and the simplified version where both agents have the same hand. Each entry is the average performance is of 20 pairs of agents that are trained with different random seeds. Cross-play score is the non-diagonal mean of each grid. Self-play score is the diagonal mean, i.e. the score attained when agents play with the peer they are trained with.

