# OpenReview forum: "Learning Intuitive Policies Using Action Features"
_ICLR.cc/2023/Conference — Submitted to ICLR 2023_

### Official Review · Reviewer_w7Kr · 2022-10-22

**Confidence:** 5
**Correctness:** 4
**Technical Novelty And Significance:** 4
**Empirical Novelty And Significance:** 4
**Recommendation:** 6

**Clarity, Quality, Novelty And Reproducibility:**

The paper is quite clearly written. I would swap section 6.1 with section 5. The paper would benefit from adding motivation for the successful architecture (SA2I), and an explanation of its behavior. It took some time for me to realize that the architecture represents the hinter OR the guesser, rather the hinter AND the guesser.

The empirical study is novel and is rigorously conducted. The findings are also novel and significant.

**Strength And Weaknesses:**

Strengths:
* This is a very interesting paper. The topic is well motivated by studies in linguistic and socio-cognitive science. I like that the authors explain intuitive concepts using well-illustrated examples.
* The problem of learning to coordinate with humans is an important and challenging problem. This study may lead to novel ways of solving this problem.
* The proposed task is easy to understand and useful in evaluating the capabilities of interest.
* The experiments are well-designed. The empirical results support the claims made by the authors.

Weaknesses:
* The authors do not offer a satisfying explanation or motivation on why the SA2I architecture produces the observed results (whereas the CA2I does not). Given that they are searching for the architecture with the right inductive bias, it is unclear what that inductive bias is? What general principles can be drawn for designing more complex architecture to tackle more challenging problems?
* The authors present the results as mere findings. I think a better way to structure the paper is to first clearly form expectations (or hypotheses) about the proposed architectures, explain why you have such expectations, and then present the results as verification of the expectations.
* The proposed task remains simplistic. It is unclear whether the obtained findings will carry over to tasks with more complex observation and action spaces.
* The method requires a common representation for the observation and the action, which may not be trivial to construct in other scenarios.


**Summary Of The Paper:**

The paper investigates whether different model architectures leverage similarities between features of the observation and the action like humans do. It proposes a communication game where the agent can reference to observation by choosing actions whose features imply features of the observation. The authors found a self-attention architecture that produces a policy that is similar to that of humans. Experiments show that the policy can coordinate successfully with humans without ever being trained with them.


**Summary Of The Review:**

I think the paper serves as a good starting point for exploring this topic. It would motivate subsequent studies on more complex tasks. I recommend acceptance.

====After Rebuttal====

I have read the authors' response. While I appreciate the explanations, my initial concerns cannot be resolved solely by the response and would require substantial revision of the paper. I thus retain my initial assessment.

---

> ### Author Response · Authors · 2022-11-16
> **Author Response**
>
> Thanks very much for your thoughtful review! We agree with the reviewer's characterizations of the strengths and weaknesses of our work. We provide additional context regarding the weaknesses below.
>
> > The authors do not offer a satisfying explanation or motivation on why the SA2I architecture produces the observed results (whereas the CA2I does not). Given that they are searching for the architecture with the right inductive bias, it is unclear what that inductive bias is? What general principles can be drawn for designing more complex architecture to tackle more challenging problems?
> > The authors present the results as mere findings. I think a better way to structure the paper is to first clearly form expectations (or hypotheses) about the proposed architectures, explain why you have such expectations, and then present the results as verification of the expectations.
>
> We agree that the submission was lacking in this respect. We believe the charcterizing different between SA2I and the other architectures is that SA2I uses the same weights to process action features and observation features. In contrast, the MLP and Attn do not consider action features at all; MLP Action In uses a different vector of weights for each input dimension; and CA2I uses a query matrix (which processes action features) that is distinct from its key and value matrices (which processes observation features). We have updated the text in the introduction, Architectures Examined section, and conclusion to reflect this hypothesis.
>
> > The proposed task remains simplistic. It is unclear whether the obtained findings will carry over to tasks with more complex observation and action spaces.
>
> We agree that the tasks are simple. However, choosing simple tasks allowed us to isolate the coordination aspect of the challenge. In more complex settings, it may be difficult to disentangle the coordination aspect from other difficult aspects. Furthermore, choosing simple tasks allowed us to perform many experimental runs. For these reasons, we feel that investigating settings with complex observation and action spaces is better left for future work.
>
> > The method requires a common representation for the observation and the action, which may not be trivial to construct in other scenarios.
>
> This is true. However, such a setting may fall outside the scope of the phenomenon we are investigating.

---

### Official Review · Reviewer_Xa5L · 2022-10-23

**Confidence:** 4
**Correctness:** 2
**Technical Novelty And Significance:** 2
**Empirical Novelty And Significance:** 2
**Recommendation:** 3

**Clarity, Quality, Novelty And Reproducibility:**

**Clarity**

As explained above, critical aspects of the work are never clearly explained. Apart from this, much of the paper is clear and well-written.

In Fig. 4, it appears that the first 6 agents chose the Sim strategy, and the last 7 agents chose the Dissim strategy. Apparently the agents were intentionally grouped together according to the strategy they chose, but I don’t see where the paper says this. Also in Fig. 4, shouldn’t the final plot be labeled SA2I instead of A2I?

**Quality**

Much of the work is carefully executed, but I cannot fairly assess the significance of the Hint-Guess results without clarifications of the points mentioned above.

**Novelty**

The Hint-Guess game appears to be new and distinctive. The architectures tested are simple and composed of very standard components.

**Reproducibility**

The eventual release of the source code will help, but won’t make up for the paper’s current lack of critical technical details.


**Strength And Weaknesses:**

**Strengths**

As motivation, the paper carefully argues for the importance of interpretable policies, in particular with respect to the exploitation of semantic relationships. As one example of such relationships, the Hint-Guess game (analogous to a type of card game) is cleverly designed to test an agent’s ability to take advantage of relationships between observations and actions. The experimental results show that SA2I is significantly more capable than the other models at this task. Additional studies and game variants characterize the gains in some detail, and even show that the best agents perform well when paired with human players.

**Weaknesses**

There would be great value in an agent architecture that demonstrated it could reliably learn interpretable policies, and one can imagine a wide variety of tasks involving semantic relationships. But no single task (such as the Hint-Guess game) by itself can provide strong evidence that a particular model’s ability to learn interpretable policies would apply to a broader set of tasks. So one major weakness is the paper’s consideration of just a single task and modest variants of it.

In addition, given the nature of the Hint-Guess game, it is hard to imagine any successful policy that did not rely on the strategy that humans find obvious here. Despite this expectation (that only one particular policy could succeed), the winning agent manages to find two winning policies, one of which (Dissim) is far less intuitive than the other (Sim). This experimental outcome raises doubt that the SA2I architecture has any general ability to learn interpretable policies.

In fact, there is a simple, alternative explanation for why attention should be expected to outperform simple MLPs on this task. In the Hint-Guess game, each observation contains unordered sets of objects (the cards in each hand). Since attention-based models (like transformers) are set processors by design, it is no surprise that applying attention to the observation would outperform an MLP. This alternative explanation calls into question the paper’s claim that the SA2I model “has a strong inductive bias toward using the relationship between actions and observations in intuitive ways.”

The term “self-play” creates confusion. The term “self-play” is commonly understood (for games like Chess or Go) to mean that an agent is playing against a copy of itself, but that does not appear to be the case in this work. For instance, “we train agents in the standard self-play setting… where the hinter and guesser are jointly trained to maximize their score”. It appears that in this paper, self-play vs. cross-play simply indicates whether an agent was trained with the (non-self) agent it is evaluated with. But it’s hard for a reader to be sure without a clearer explanation. Similarly, in the intra-AXP setting, what is the algorithm being shared?

Several crucial aspects of the Hint-Guess experiments are never adequately explained. For instance, the hinter and guesser are apparently two separate agents. But is one agent trained to always be the hinter, while the other agent is always the guesser? If so, do the hinter and guesser share any weights? Or is each agent trained in both roles, sometimes as the hinter and sometimes as the guesser? If so, then how does the agent know in any particular game whether it is playing as the hinter or as the guesser? Is this information included in the observation?

Important details of the observations are left out. For instance, “the observation input is a sequence of card representations for both hands $H_1$ and $H_2$, as well as the representation of the target card, $C_i^2$ (for the hinter) or the hinted card $C_j^1$ (for the guesser)”. Since the hand size is 5 for the main experiments, this could mean that each agent will see 11 cards at once:  2 hands plus a copy of either the target or the hinted card. Or are only 10 cards visible, and the target or hinted card is specified some other way? And how does the agent know which cards are its own? Perhaps the first 5 cards are always the agent’s own cards, or maybe some flag or embedding is applied to indicate ownership. Or is this information conveyed some other way?

The block diagrams in Fig. 3 are helpful, but they also raise important unanswered questions. When multiple Q values are output, how are those values mapped to specific cards, given that the order of the cards is randomly permuted on input, and the Reduce Mean operation loses order information? Does each Reduce Mean operation average a set of input vectors to produce a vector of the same dimensionality as the inputs? And exactly which vectors serve as the inputs?

Finally, since this work proposes a novel task, the baseline results would be more meaningful if the baseline model hyperparameters had been tuned. But hyperparameter tuning is never mentioned.


**Summary Of The Paper:**

This work investigates the learning of intuitive, interpretable policies that leverage features shared between observations and actions, by means of a novel two-player game (Hint-Guess) where such relationships play a pivotal role. Several different model architectures are evaluated, and they are found to demonstrate widely varying abilities. One particular attention architecture (SA2I) far outperforms the others on this task, and its policies are shown to take advantage of the task structure in ways that one would intuitively expect.

**Summary Of The Review:**

The paper makes an interesting claim, but does not provide enough evidence to support it.

==== After discussions with authors ====

The authors have supplied many important details that were missing from the original paper. But in my view, the most serious limitation of the work continues to be its reliance on a single, particularly simple task of no practical importance. This limitation was also called out by the other reviewers.

In addition, I still cannot see anything particularly surprising about the main results on this task, presented in Table 1. As noted by the paper, and confirmed by the authors in our extensive discussion, the poor performance of MLP and Attn is easily explained by the fact that they “do not consider action features at all”, which are crucial factors in this task by construction. The only remaining question is whether self-attention (SA2I) or cross-attention (CA2I) is more appropriate to the task. Since cross-attention is a less-costly special case of self-attention, it is no surprise that self-attention has an advantage. (The cross-play setting is the important one, as it factors out the possibility of learning a private language.)

For these reasons, and despite the commendable work that this paper represents, I have to leave my score unchanged. I look forward to extensions of this work on the important connection between semantic relationships and interpretable policies.

---

> ### Author Response · Authors · 2022-11-16
> **Author Response (1/2)**
>
> Thanks very much for your thoughtful review! We agree with the reviewer that we inadvertantly omitted some experimental details, which we have now added to the paper. We thank the reviewer for their diligence in observing this and hope that the rebuttal below and the changes we made to the paper address these concerns.
>
> > There would be great value in an agent architecture that demonstrated it could reliably learn interpretable policies, and one can imagine a wide variety of tasks involving semantic relationships. But no single task (such as the Hint-Guess game) by itself can provide strong evidence that a particular model’s ability to learn interpretable policies would apply to a broader set of tasks. So one major weakness is the paper’s consideration of just a single task and modest variants of it.
>
> We agree that this would be of great value. However, as noted by the reviewer, such a thing is difficult to show definitively. In the submission we chose to thoroughly investiate a class of tasks that isolates coordination problems in partially observable settings. While we agree that this does not definitively show our result in generality, we feel that our results on simple coordination tasks are interesting and of value to the community.
>
> > In addition, given the nature of the Hint-Guess game, it is hard to imagine any successful policy that did not rely on the strategy that humans find obvious here.
>
> **Our experiments show that most architectures find successful strategies that are not obvious to humans.** For these results, please see Table 1. This table shows that MLP, MLP Action In, Attn, and CA2I all discover strategies that perform as well or better than the strategies that are interpretable to humans (i.e., sim and dissim).
>
> > In fact, there is a simple, alternative explanation for why attention should be expected to outperform simple MLPs on this task. In the Hint-Guess game, each observation contains unordered sets of objects (the cards in each hand). Since attention-based models (like transformers) are set processors by design, it is no surprise that applying attention to the observation would outperform an MLP. This alternative explanation calls into question the paper’s claim that the SA2I model “has a strong inductive bias toward using the relationship between actions and observations in intuitive ways.”
>
> **Our experiments falsify this alternative explanation.** First, as stated in the experimental setup section, we permute the hands of the MLP-based architectures, thereby preventing them from learning order-specific conventions. Second, and perhaps more importantly, our experiments show that Attn and CA2I, which are attention-based models, do not posssess the same inductive bias that SA2I possesses -- i.e., they do not tend to learn interpretable polices.
>
> > The term “self-play” creates confusion. The term “self-play” is commonly understood (for games like Chess or Go) to mean that an agent is playing against a copy of itself, but that does not appear to be the case in this work. For instance, “we train agents in the standard self-play setting… where the hinter and guesser are jointly trained to maximize their score”. It appears that in this paper, self-play vs. cross-play simply indicates whether an agent was trained with the (non-self) agent it is evaluated with. But it’s hard for a reader to be sure without a clearer explanation.
>
> We agree with the reviewer that the classical usage of "self-play" means that an agent is playing against a copy of itself. We also agree that that is not the case in this work. The way we use the term is in line with recent literature (e.g., Improving Policies via Search in Cooperative Partially Observable Games; Lerer et al. 2020, Hu et al. 2020, Hu et al. 2021), who use the term self-play to refer to a setting in which agents are evaluated with the teammates with whom they were trained, as is suggested by the reviewer. Note that we define this usage in the submission: "The SP score is the diagonal mean, i.e. the score attained when agents play with the peer they are trained with." Please let us know if you feel additional explanation is meritted and we are happy to add it.
>
> > Similarly, in the intra-AXP setting, what is the algorithm being shared?
>
> The algorithm being shared is the algorithm being evaluated. We describe a procedure for estimating intra-AXP for a particular algorithm below for clarity:
>
> input: algorithm $A$
> - $scores = []$
> - for many runs:
>     -    $teams = []$
>     -    for number of teammates:
>           -   train team $x'$ using algorithm $A$
>           -   $teams.append(x')$
>     - make team $x$ consisting of one member of each team in $teams$
>     - $scores.append(evaluate(x))$
> - return $avg(scores)$

---

> > ### Author Response · Authors · 2022-11-16
> > **Author Response (1/2)**
> >
> > > Several crucial aspects of the Hint-Guess experiments are never adequately explained. For instance, the hinter and guesser are apparently two separate agents. But is one agent trained to always be the hinter, while the other agent is always the guesser? Or is each agent trained in both roles, sometimes as the hinter and sometimes as the guesser? If so, then how does the agent know in any particular game whether it is playing as the hinter or as the guesser? Is this information included in the observation?
> >
> > One agent is always trained to be the hinter and one agent is always trained to be the guesser. We have added text to the experimental section to clarify this.
> >
> > > Important details of the observations are left out. For instance, “the observation input is a sequence of card representations for both hands $H_1$ and $H_2$, as well as the representation of the target card, $C_i^2$ (for the hinter) or the hinted card $C_j^1$ (for the guesser)”. Since the hand size is 5 for the main experiments, this could mean that each agent will see 11 cards at once: 2 hands plus a copy of either the target or the hinted card. Or are only 10 cards visible, and the target or hinted card is specified some other way?
> >
> > Each agent sees 11 cards at once.
> >
> > > And how does the agent know which cards are its own? Perhaps the first 5 cards are always the agent’s own cards, or maybe some flag or embedding is applied to indicate ownership. Or is this information conveyed some other way?
> >
> > There is a binary feature added to each card's featurization specifying whether or not it belongs to the agent. We have added text to clarify this point.
> >
> > > The block diagrams in Fig. 3 are helpful, but they also raise important unanswered questions. When multiple Q values are output, how are those values mapped to specific cards, given that the order of the cards is randomly permuted on input, and the Reduce Mean operation loses order information?
> >
> > For the architectures that do not take action features as input, the actions are semantically of the form "hint card with color X and number Y" and "guess card with color X and number Y". We mask the output so that only legal actions (i.e., actions that correspond to a card in the players hand) can be selected. We have added text to the submission to clarify this point.
> >
> > > Does each Reduce Mean operation average a set of input vectors to produce a vector of the same dimensionality as the inputs? And exactly which vectors serve as the inputs?
> >
> > The reduce mean operation averages $S$ number of vectors of dimension $E$ and produces a vector of length $E$, following the naming conventions of PyTorch documentation: https://pytorch.org/docs/stable/generated/torch.nn.Transformer.html.
> > The set of vectors output by the attention block serve as input.
> >
> > > In Fig. 4, it appears that the first 6 agents chose the Sim strategy, and the last 7 agents chose the Dissim strategy. Apparently the agents were intentionally grouped together according to the strategy they chose, but I don’t see where the paper says this.
> >
> > Good point -- yes, we used a clustering algorithm to visualize the results. We added some text to the paper to clarify this.
> >
> > > Also in Fig. 4, shouldn’t the final plot be labeled SA2I instead of A2I?
> >
> > Good catch -- fixed.
> >
> > > The eventual release of the source code will help
> >
> > **We commit to releasing source code if accepted.**
> >
> > ---
> >
> > We thank the reviewer for their attention to detail and hope that we can continue to have constructive discussion about the submission.

---

> > > ### Comment · Reviewer_Xa5L · 2022-11-18
> > > **Thank you**
> > >
> > > Thank you for providing so many additional details!
> > >
> > > Regarding this point:
> > >
> > > **“we permute the hands of the MLP-based architectures, thereby preventing them from learning order-specific conventions”**
> > >
> > > If I understand what you are saying here, then it dovetails with the point I was making, which is that MLPs don’t handle permutations of input sub-vectors well. Or are you trying to make another point? Can you explain exactly what is being permuted here? Are you changing the order of sub-vectors within each vector passed to the MLP?

---

> > > > ### Author Response · Authors · 2022-11-18
> > > > **Thanks for your engagement!**
> > > >
> > > > Ah, we understand now, thanks for your clarifications. Yes -- we were permuting the input sub-vectors as the reviewer suggests. However, we have also performed experiments in which the MLP receives the hands without permutation -- **we found that this permutation-sensitive version of the MLP performs similarly to the permutation-insensitive MLP results reported in the paper.** We chose to include the permutation-insensitive version rather than the permutation-sensitive version for an apples-to-apples comparison with the set-based architectures. We are happy to add the permutation-sensitive MLP results to the appendix, if the reviewer would find that interesting. Please let us know :)

---

> > > > > ### Comment · Reviewer_Xa5L · 2022-11-18
> > > > > **Permutations**
> > > > >
> > > > > No need to add the un-permutated MLP results to the appendix. It's expected that these results would be similar to the permuted results. In either case, the order of the cards in a hand is irrelevant. This irrelevance fits the inductive bias of self-attention (being permutation-equivariant), but MLPs don't share this bias. To me, this still seems like the simplest explanation for the under-performance of MLP models in the Cross-Play setting. Meanwhile, their good performance in the Self-Play setting seems likely to be due to learning a private language, as your paper hypothesizes. Please correct me if I'm missing something.

---

> > > > > > ### Author Response · Authors · 2022-11-18
> > > > > > **Re: Explanation**
> > > > > >
> > > > > > Thanks again for your replies!
> > > > > >
> > > > > > We agree that that this may be *one reason* that MLPs perform poorly in the cross-play setting. However, our results show that **some attention-based architectures (Attn and CA2I) perform equally poorly as MLPs in the cross play setting**; so we feel that the idea that *attention outperforms MLPs because they're set processors by design* is unable to fully explain the results from experiments. Indeed, it is only the self-attention architecture that processes both the observation features and the action features that performs well. Our hypothesis for this phenomenon is that it is important to use the same parameters to process action features and observation features. Attn cannot do this because it does not use action features; CA2I cannot do this because the query parameters are distinct from the key and value parameters. However, there are other set-based architectures (e.g., deep sets) that may be able to do this. We would not be able to complete experiments investigating this before the update deadline but are willing to commit to having them in the next revision if the reviewer would find them interesting.
> > > > > >
> > > > > > **To the extent that the reviewer is suggesting that our takeaway is something along the lines of *set-based architectures have a better inductive bias for learning intuitive policies when the featurized action is included as part of the set*, we agree!** To the extent that the reviewer is suggesting that this should've been expected, we agree that it seemed plausible (hence why we investigated it in the first place). Still, we found the strength of this inductive bias surprising: We did not anticipate the level of structure and interpretability of the policies produced by running simple independent Q-learning on top of SA2I. And, even though it is a simple game, the fact that SA2I Sim achieved human level ad hoc coordination from self-play training was impressive to us.
> > > > > >
> > > > > > Please let us know if we are misinterpreting anything!

---

> > > > > > > ### Comment · Reviewer_Xa5L · 2022-11-18
> > > > > > > **Action features**
> > > > > > >
> > > > > > > In another response, you said that unlike SA2I, "MLP and Attn do not consider action features at all". For this task, it certainly does seem important to consider action features. So is this a sufficient explanation for SA2I's good performance?

---

> > > > > > > > ### Author Response · Authors · 2022-11-18
> > > > > > > > **Thanks again for your continued replies!**
> > > > > > > >
> > > > > > > > > In another response, you said that unlike SA2I, "MLP and Attn do not consider action features at all".
> > > > > > > >
> > > > > > > > Indeed, this is correct.
> > > > > > > >
> > > > > > > > > For this task, it certainly does seem important to consider action features.
> > > > > > > >
> > > > > > > > We agree!
> > > > > > > >
> > > > > > > > > So is this a sufficient explanation for SA2I's good performance?
> > > > > > > >
> > > > > > > > We would still maintain that this does not fully explain the results: **Our CA2I architecture 1) considers action features and 2) is a set processor attention-based architecture. Yet, it performs similarly poorly to MLP, MLP Action In, and Attn in cross-play performance.** We see this as evidence of the importance of using the same parameters to process both the action features and observation features, as SA2I does. (In contrast, in its first layer, CA2I only uses the query parameters to process the action features, and only uses the key and value parameters to process the observation features --- ie, it uses cross attention instead of self attention.)
> > > > > > > >
> > > > > > > > Thanks again for your continued engagement! Please let us know if we can provide any additional clarification or if we are misunderstanding anything the reviewer is saying.

---

### Official Review · Reviewer_rXa8 · 2022-10-25

**Confidence:** 3
**Correctness:** 3
**Technical Novelty And Significance:** 4
**Empirical Novelty And Significance:** 4
**Recommendation:** 6

**Clarity, Quality, Novelty And Reproducibility:**

* The paper is well-motivated and clearly written.

* In Figure 4 and Figure 6, "A2I" needs to be "SA2I".

* The proposed problem and experimental setup are novel.

* The experiments are well-organized and thoroughly analyzed. But, the experiments are done in a simplified task, which makes the conclusion weaker.


**Strength And Weaknesses:**


### Strengths

* This paper points out an interesting perspective of learning shared action-observation features for coordination that can be used commonly across independently trained agents.

* The investigation of multiple different Q-network architectures shows clear evidence of the necessity for action representations in the cross-play experiments.

* The evaluation of human subjects further proves that the learned strategies are compatible with humans.


### Weaknesses

* The problem formulation in the paper suggests a novel and interesting research direction. However, the example tasks seem to be too simple and not well connected to the realistic or practical use of the emerged (shared) action representations. Thus, one thing that can further strengthen the paper would be additional discussion or experiments with more realistic or practical scenarios.

* The experiments are conducted on the one-feature scenarios and two-feature scenarios. It might be interesting to see whether the findings from these scenarios hold for a more number of features. It might be also interesting to discuss whether the same hypothesis holds for more than two agents.

**Summary Of The Paper:**

This paper suggests an interesting idea of real-world actions often being grounded by observations and the shared action-observation features naturally emerging for coordination between agents. This paper hypothesizes that the policy architecture is crucial for emerging the shared action-observation features. The experiments are done using the simple hint-guess game and demonstrate that the sharable action-observation features emerge only when each action is fed into the Q-network.

**Summary Of The Review:**

This paper proposes an interesting and potentially important problem for the community. However, the evaluation is done in too simplified tasks, which makes the paper not very convincing. I still value the novelty of the paper, so leaning toward weak acceptance. A few more tasks with sufficient complexity would make this paper very strong.

---

> ### Author Response · Authors · 2022-11-16
> **Author Response**
>
>
> Thanks very much for your thoughtful review! We agree with the reviewer's characterizations of the strengths and weaknesses of our work. We provide additional context regarding the weaknesses below.
>
> > the example tasks seem to be too simple and not well connected to the realistic or practical use of the emerged (shared) action representations. Thus, one thing that can further strengthen the paper would be additional discussion or experiments with more realistic or practical scenarios.
>
> We agree that the tasks are simple. However, choosing simple tasks allowed us to isolate the coordination aspect of the challenge. In more complex settings, it may be difficult to disentangle the coordination aspect from other difficult aspects. Also, choosing simple tasks allowed us to perform many more experimental runs than we would've been able to afford on harder tasks. For these reasons, we feel that investigating complex settings is better left for future work.
>
> > The experiments are conducted on the one-feature scenarios and two-feature scenarios. It might be interesting to see whether the findings from these scenarios hold for a more number of features.
>
> We agree that this would be interesting. We will not be able to complete such an experiment prior to the end of the rebuttal period, but commit **commit to having such an experiment in the camera ready version if accepted.**
>
> > In Figure 4 and Figure 6, "A2I" needs to be "SA2I".
>
> Thanks for this catch! Fixed.

---

> > ### Comment · Reviewer_rXa8 · 2022-11-17
> > **Thank you for the response**
> >
> > Thank you for your response!
> >
> > I understand that simplified tasks can help diagnose different approaches in diverse aspects, and make experiments efficient. However, it would still be valuable to discuss potential (realistic) applications and scalability of the proposed approach in this paper.

---

### Decision · Program_Chairs · 2023-01-20

**Decision:**

Reject

**Justification For Why Not Higher Score:**

All reviewers point our the same limitation of having a single the very simplified task and the arising question if the findings will generalize, which cannot be addressed without a substantive re-write.
Having a single task is understandable as it allows exploring the method in detail, but combined with the 'non-surprising' outcomes the paper is a bit too limited in its current form.

**Justification For Why Not Lower Score:**

N/A

**Metareview: Summary, Strengths And Weaknesses:**

Summary:
The paper explores the emergence of shared features for observations and actions in a multi-agent coordination setting. That this is indeed the case is demonstrated on a single task where multiple variants of agents are explored.

Strengths:
- A interesting new perspective on multi-agent learning
- Exploration of architectures
- Experiments with humans

Weaknesses:
- The paper only focuses on a single task and it remains unclear how general the method is
- There are still doubts about how fair the comparison is, given that the baselines don't have access to features that are crucial for solving the task
- Requires constructing a common representation

**Summary Of Ac-Reviewer Meeting:**

N/A